# Mechanosensitive channel YnaI has lipid-bound extended sensor paddles

Wenxin Hu [1], Zhiming Wang[1] & Hongjin Zheng [1✉]

The general mechanism of bacterial mechanosensitive channels (MS) has been characterized by extensive studies on a small conductance channel MscS from *Escherichia coli (E. coli)*. However, recent structural studies on the same channel have revealed controversial roles of various channel-bound lipids in channel gating. To better understand bacterial MscS-like channels, it is necessary to characterize homologs other than MscS. Here, we describe the structure of YnaI, one of the closest MscS homologs in *E. coli*, in its non-conducting state at 3.3 Å resolution determined by cryo electron microscopy. Our structure revealed the intact membrane sensor paddle domain in YnaI, which was stabilized by functionally important residues H43, Q46, Y50 and K93. In the pockets between sensor paddles, there were clear lipid densities that interact strongly with residues Q100 and R120. These lipids were a mixture of natural lipids but may be enriched in cardiolipin and phosphatidylserine. In addition, residues along the ion-conducting pathway and responsible for the heptameric assembly were discussed. Together with biochemical experiments and mutagenesis studies, our results provide strong support for the idea that the pocket lipids are functionally important for mechanosensitive channels.

[1] Department of Biochemistry and Molecular Genetics, University of Colorado Anschutz Medical Campus, School of Medicine, Aurora, USA.
✉email: hongjin.zheng@cuanschutz.edu

Osmotic balance is essential to all cell survival. When experiencing a hypo-osmotic shock (lower osmolyte concentration outside), cells swell quickly with water flooding in. This physical swelling causes enormous turgor pressure against the cellular membrane. To maintain the membrane integrity as well as a functional metabolism inside the cells, cytosolic solutes, mostly ions, are quickly released through the open pores of mechanosensitive (MS) channels. In bacteria, two families of MS channels[1,2] have been identified: MscL and MscS. In general, MscL proteins have a large conductance (such as ~3nS for *Escherichia coli* MscL), while MscS proteins have a small conductance (such as ~1nS for *Escherichia coli* MscS). Several structures of closed and open states of MscS from different species[3–6] have been determined, together with biochemical and electrophysiological data, leading to an emerged working model under the concept of "Force-From-Lipids"[7,8]: in lipids bilayers, integral membrane proteins, such as MS channels, are surrounded by annular lipids and thus gated by the force transferred from those lipids. For example, MscS rests in the closed state with lipids bounds to multiple locations including the central pore (pore lipids), the pockets between paddle domains (pocket lipids), as well as flexible loops between transmembrane helices (hook lipids)[9,10]. When membrane tension applies, the channel opens by moving transmembrane helices, which are the results of moving corresponding protein-bound lipids into the bilayer. Although which lipids are more relevant to the pressure sensing is debatable, the functional importance of these lipids is in consensus[6,10,11].

MscS family members are found in bacteria, archaea, fungi, and plants, but not in animals, suggesting that they may serve as pharmaceutical targets to treat pathogenic infections in humans[12,13]. MscS homologs have diverse sequences, containing 3–11 transmembrane helices (TMs), a large cytosolic domain, and sometimes a periplasmic domain[2,14]. It remains unclear how different bacterial MscS homologs perform similar function with such diverged sequences. One of MscS closest homolog is YnaI from *Escherichia coli*. A recent publication has revealed high-resolution structures of YnaI showing a possible gating mechanism based on its flexible pore helices[15]. However, the interactions between YnaI and its surrounding lipids are not fully understood. To shed light on this, we have determined the high-resolution structure of YnaI in the closed state at 3.3 Å resolution by cryo electron microscopy (cryo-EM). With this greatly improved structure, it becomes clear that the five TMs of YnaI form extended sensor paddles with pockets in between, which are comparably larger than their counterparts in MscS that has only three TMs. In addition, there are clear lipid densities within those pockets, which have been shown critical for mechanosensing in MscS[6]. However, recent MscS structures[10] contradict previous notion by showing that these pocket lipids are positionally too far away from the lipid bilayer to be functionally important. In this study, we have performed biochemical analysis on the freshly revealed lipid-binding residues in the YnaI pockets, so to provide valuable insights into the argument about general mechanosensing mechanisms of MscS-like channels.

## Results

**Overall architecture of YnaI.** We overexpressed the full-length YnaI and purified it in detergent lauryl maltose neopentyl glycol (LMNG). The results show a homogenous single peak for the heptameric YnaI on gel filtration chromatography and one clear band on SDS-PAGE (Fig. S1). We first examined the sample by negative stain EM. The results show that YnaI are well separated in solution and appears to be homogenous in size. Two-dimensional (2D) averages of the stained YnaI particles already show a channel in the middle of the molecule and a seven-fold symmetry along the channel (Fig. S2). However, when analyzed under cryo conditions, YnaI adopts preferred orientations in the vitrified ice, presenting mostly top/bottom views as shown in the 2D averages (Fig. S3). Here we define the top view as looking from the periplasmic side down the channel, the bottom view as looking from the cytosolic side up through the channel, while the side views are tilted views in between the top and bottom views. With limited side view particles, we attempted reconstruction and, not surprisingly, obtained cryo-EM maps with distorted densities that are not interpretable. To overcome this problem for high-resolution structure determination, we tried various methods including tilting the grid, using continuous carbon, as well as adding different detergents. The best result is obtained by adding fluorinated Fos-Choline-8 to the sample right before freezing. As shown in Fig. S3, various side views of YnaI particles are present in the cryo images, and features of secondary structures of YnaI are clearly visible in corresponding 2D averages. After data processing in RELION[16], we obtained a final reconstruction of YnaI at an overall resolution of 3.3 Å (Table 1, Fig. S4).

YnaI forms a heptamer with a dimension of 130 Å x 100 Å x 100 Å in space (Fig. 1). In each YnaI monomer, the transmembrane domain contains five TMs (Fig. S5), named −2, −1, 1, 2, and 3 starting from the N-termini here. TM1 ~3 are structurally homologues to the three TMs in MscS[3], which are well conserved

**Table 1 Cryo-EM data collection, refinement, and validation statistics.**

|  | YnaI (EMDB-20862) (PDB 6URT) |
|---|---|
| **Data collection and processing** | |
| Magnification | 45,000 |
| Voltage (kV) | 200 |
| Electron exposure (e–/Å²) | 64.3 |
| Defocus range (μm) | −0.1 to −2.5 |
| Pixel size (Å) | 0.864 |
| Symmetry imposed | C7 |
| Initial particle images (no.) | ~800,000 |
| Final particle images (no.) | 177,979 |
| Map resolution (Å) | 3.3 |
| FSC threshold | 0.143 |
| Map resolution range (Å) | 3.0–4.6 |
| **Refinement** | |
| Initial model used (PDB code) | N/A |
| Model resolution (Å) | 3.3 |
| FSC threshold | 0.5 |
| Model resolution range (Å) | |
| Map sharpening *B* factor (Å²) | −129.9 |
| Model composition | |
| Non-hydrogen atoms | 17920 |
| Protein residues | 2317 |
| Ligands | |
| *B* factors (Å²) | |
| Protein | −120 |
| Ligand | |
| R.m.s. deviations | |
| Bond lengths (Å) | 0.009 |
| Bond angles (°) | 1.050 |
| Validation | |
| MolProbity score | 2.85 |
| Clashscore | 6.16 |
| Poor rotamers (%) | 0.7 |
| Ramachandran plot | |
| Favored (%) | 91.14 |
| Allowed (%) | 8.86 |
| Disallowed (%) | 0 |

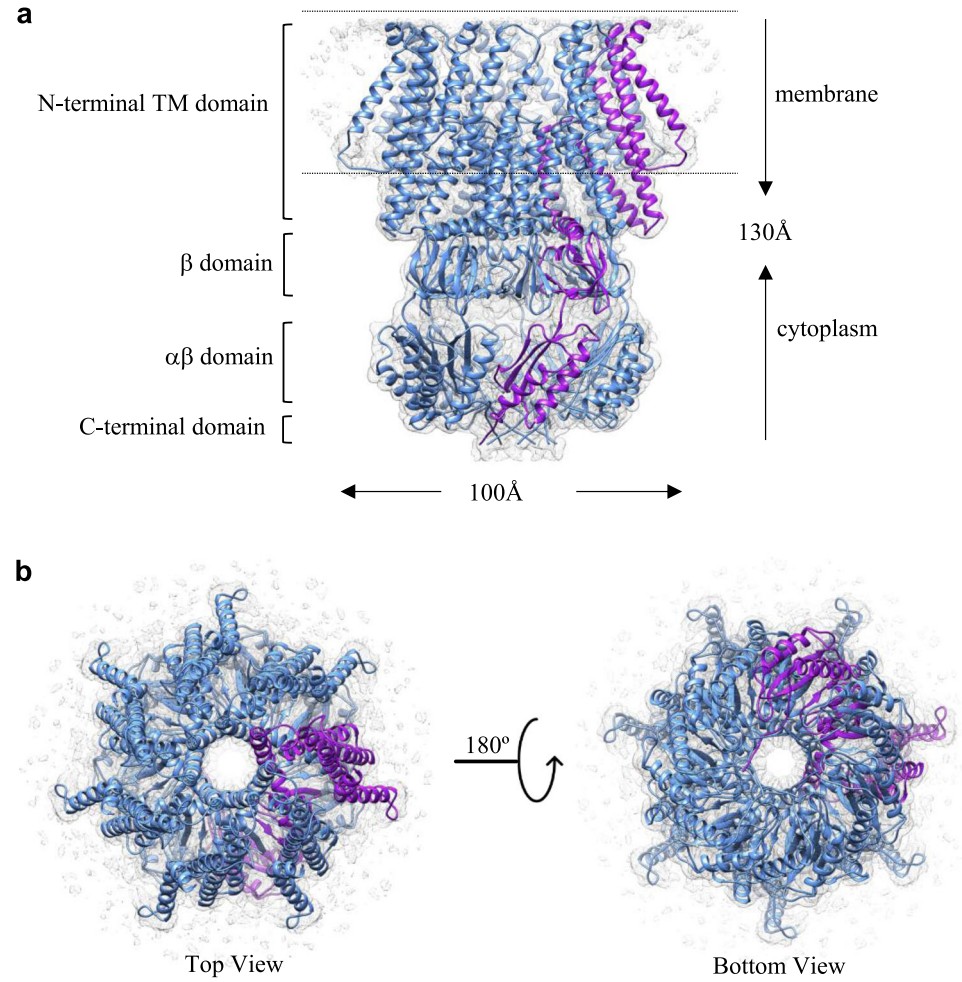

**Fig. 1 Overall architecture of YnaI. a** Side view of the atomic model of heptameric YnaI (blue with one subunit in purple) fits into its cryo-EM density map (gray). **b** Top (left) and bottom (right) views of YnaI.

among MscS family members. Thus, this nomenclature is ought to make it easier to directly compare with known MscS structures. In YnaI, the first four TMs (TM-2 ~2) form an extended large sensor paddle, while the last one (TM3) lies along the central ion conducting channel. The cytoplasmic portion of YnaI is similar to MscS and can be divided into three domains from top to bottom: β domain, αβ domain, and the C-terminal domain. Right below the TM3 is the β domain, which has ~60 residues and contains a twisted β sheet consists of five short β strands. The αβ domain has ~90 residues, composed of two α helices and three β strands. The last ~20 residues make up the C-terminal domain, which is predicted to form a β-barrel. In our YnaI structure, the C-terminal domain indeed forms a β-barrel-like structure, but each subunit is more like an extended loop instead of a standard β strand.

**Extended paddle domain**. YnaI is the closest homolog to MscS with the major difference in the TMD. The high-resolution structures of MscS have been determined in closed (non-conducting)[3] and open (conducting)[4,6,17] states in detergents, as well as in closed state in lipid nanodiscs[10,18]. All the high-resolution structures of MscS indicate that TM1 and TM2 together form a paddle. This paddle is tilted away from the channel axis and is responsible for sensing the tension within the membrane bilayer. The exact arrangement of N-terminal residues (~20) before TM1 in MscS has recently been observed in

nanodiscs, revealing a novel membrane-anchoring fold that is important for channel activation[10,18]. A previous YnaI structure has partially revealed its paddle domain from TM1 to TM3, which can be well aligned with the non-conducting MscS[19]. Here, in our structure of YnaI, the organization of the complete paddle domain is revealed. It is worth noting that the local resolution of TM-2 in the paddle is only 4–4.5 Å, which is good enough to trace backbones of the helix but not side chains (Fig. S5). Thus, in the final model, we designated all residues in TM-2 (residues 4–30) as Alanine. To assess the conformation of YnaI, we superimposed our model with all available MscS structures. We found that YnaI best matches with the closed MscS structures, such as 6PWN[10], suggesting that our YnaI structure is also in a closed state (Fig. 2a). Specifically, TM1 and TM2 form a helix bundle in YnaI, which is equivalent to the TM1 and TM2 bundle in MscS. TM-2 and -1 in YnaI appears to be another helix bundle that seems a duplicate of the bundle of TM1 ~2. In addition, TM-2 ~−1 bundle extends towards the surrounding lipids the same direction as TM1 ~2. This leaves no MscS-like N-terminal membrane-anchoring fold in YnaI. We examined the interactions between the two bundles, which is the interface between TM-1 and TM1. Not surprisingly, they are associated together mainly through hydrophobic packing. But, several charged residues within close proximity may also contribute to the directional stability of the extended paddle: K93 in TM1 as well as H43, Q46, and Y50 in TM-1 (Fig. 2b). To assess whether the bundle stability is functionally important, we mutated these residues to Ala and

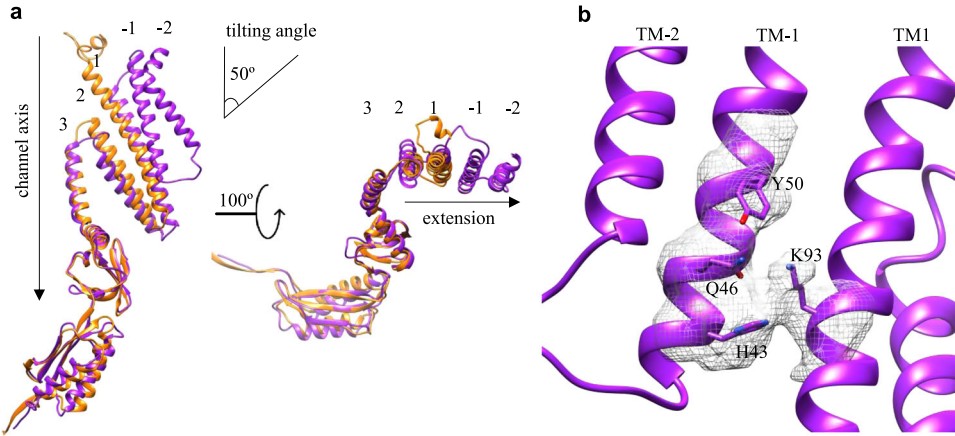

**Fig. 2 Extended tilting paddle domain in YnaI. a** Monomers of YnaI (purple) and MscS (PDB: 6PWN, orange) are superimposed. The paddle containing TM-2 ~2 is tilted relative to the vertical channel axis (left) and all four TMs are extended in the same direction into the bilayer (right). **b** Specific polar residues that contribute to the stability of the paddle domain and their experimental densities (dark gray).

made two more mutants ΔN70 (deleting TM-2 and TM-1) and ΔN32 (deleting TM-2). We performed osmotic downshock experiments as established in studies of other MS channels[2,20]. To do that, we generated an *E. coli* strain (MS-1106) by deleting six MS channel genes (five MscS members and MscL)[21], followed by λDE3 lysogenization. We noticed that all mutants, except ΔN70, express as good as YnaI wild type (Fig. S6a). During an osmotic down shock from 500 mM to ~170 mM salt (Fig. S6b), the survival rate of MS-1106 (DE3) is below 1%. Here, as a positive control, we expressed MscS in the MS-1106 strain and the survival rate is ~80%. While with the expression of YnaI, the survival rate improves to ~50%. However, all the mutants' ability to protect the cells against osmotic shock were decreased comparing to the wild type, suggesting that the paddle stability is important for YnaI to function properly.

The tilting angle of TM-2 ~−1 bundle relative to the vertical channel axis is the same as the tilting angle of TM1 ~2 bundle, ~50° (Fig. 2a). This tilting of TM bundles is similar to MscS[10], suggesting that the membrane bilayer surrounding YnaI is curved and the curvature is probably the same as MscS. The bottom one thirds of the TM1 ~2 bundle is exposed to the cytosol and only the top two thirds are in the membrane, while the TM-2 ~−1 bundle is completely buried in the bilayer. In addition, it is noticeable that the TM-2 ~−1 bundle is physically elevated along the normal of the membrane bilayer comparing to the TM1 ~2 bundle. The elevation and extension of TM1 ~2 makes the area of the curved lipid bilayer around YnaI much larger than that around MscS in closed state[3,9,10]. This larger curved area may contribute to the fact that YnaI channel opens at ~1.5 times higher pressure comparing to MscS[22].

**Lipid pockets in between the paddles.** In YnaI, TM3 is the central helix with a kink around the position G160 (Fig. 3a, S5). Specifically, residues L142–A159 form the pore lining helix TM3a, while residues K161–D176 form the cytosolic helix TM3b that points outward of the central pore, aligns almost parallel to the membrane plane, and connects to the rest of the cytosolic domains. When looking at the membrane domain of the YnaI structure from the side towards the central channel, there appears to be seven cone-shaped pockets, each of which is formed by three adjacent subunits. The pocket between subunits A (purple) and B (blue) can be described as follows (Fig. 3a): two TM3b from subunits A and G (cyan) join together to form the flat base of the

cone, two TM3a from subunits A and B form the back wall, two sensor paddles from subunits A and B form the side walls, while the front of the cone is open to the membrane bilayer. Interestingly, there are clear EM densities within the seven pockets, which are presumably co-purified lipids. To confirm that, we performed matrix-assisted laser desorption/ionization mass spectrometry (MALDI-MS), and found peaks within the size range of 500–800 Da, corresponding well with lipid molecules. To identify what types of lipids they are, we performed thin-layer chromatography (TLC) with purified YnaI that goes through gel filtration twice. The result shows that the co-purified lipids are actually a mixture (Fig. S7) of cardiolipin (CL), phosphatidylethanolamine (PE), phosphatidylglycerol (PG), as well as phosphatidylserine (PS). Among them, CL, PE, and PG are predominant lipids in *E. coli*[23]. By comparing with the standard *E. coli* total lipid extract, it seems that the co-purified lipid mixture is particularly enriched in PS and CL. These results suggest that the pockets in YnaI are surely for lipids, but not for one specific type. Instead, it is big enough to accommodate multiple lipid molecules of mixed types. However, the observed density is only good for one lipid molecule, probably because other lipids are averaged out during reconstruction due to the heterogeneity. The bulky lipid density (Fig. 3b) is in between the following two residues: R120 in TM2 of subunit B and Q100 in TM1 of subunit A. In addition, the loop between TM-2 and TM-1 of the subunit A is close to the lipid density. The EM density of this loop is not well resolved, probably because it contains many flexible and positively charged residues (31-RRGNRKRKG-39). We speculate that the lipid polar headgroup is surrounded by R120, Q100 and the loop. This is supported by the TLC results that R120A mutant binds less lipids overall comparing to the wild type (Fig. S7). As for the lipid tail, one clearly extends upwards, while another seems to extend deeper into the pocket, but its density is not very clear. This lipid arrangement is comparable to the lipids found in MscS[9,11,18].

The conformation and organization of the cone-shaped pockets in our YnaI structure are similar to that found in MscS at the closed state but are much bigger in size, because of the extended paddle domains in YnaI. Collective results from molecular dynamics simulation, mutagenesis, fluorescence quenching, structural biology, as well as other biophysical methods have demonstrated that there are lipids constantly going in and out of the MscS pockets[24]. The positional arrangement of these lipid molecules[6,18] is the same as what we observed in YnaI: with the headgroup close to the open edge and the tails stick into the pockets. Specifically, hydrophobic

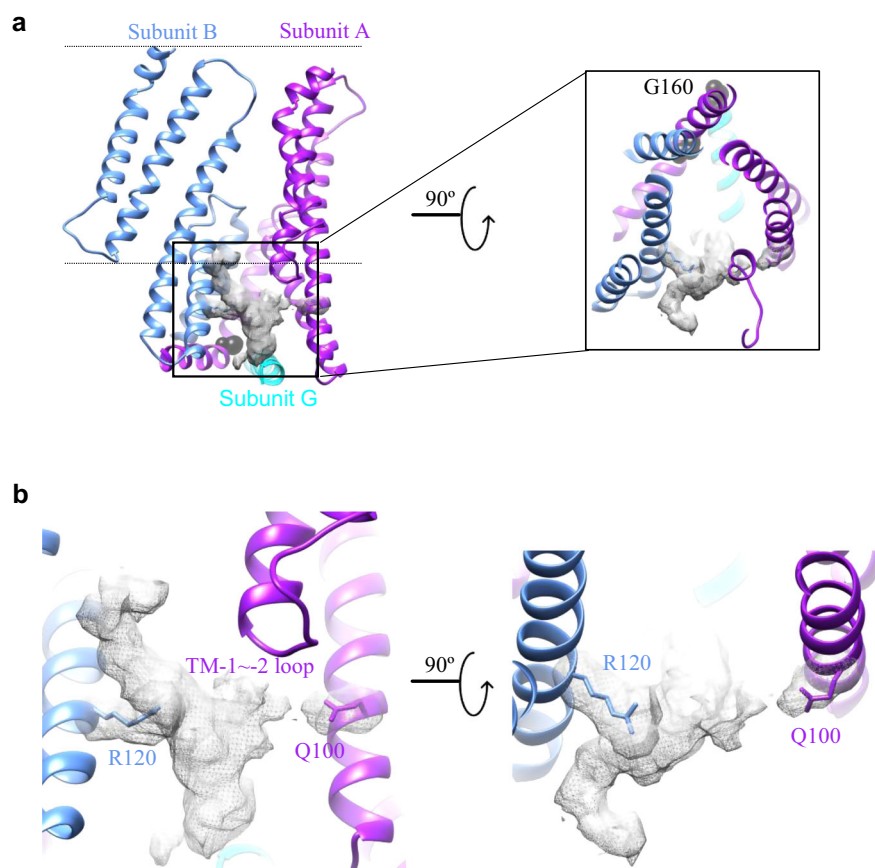

**Fig. 3 Lipids accommodated in the pockets between paddle domains. a** The cone-shaped pocket formed by three adjacent paddle domains (blue, purple, and cyan) in the membrane (indicated by dashed lines). Within the pocket, there is a bulky lipid density (gray mesh). The kink point in TM3, G160, is shown as black balls. **b** Zoomed-in views of the lipid density (as boxed in a) in the pocket. R120 and Q100 seem to directly interact with the lipid headgroup.

residues deep in the pockets, such as A103, V107, L111, and A119 (equivalent to G152, V156, G160, and F168 in YnaI) are close to the lipid tails, while F127, I150, and F151 (equivalent to D176, W201, and R202 in YnaI) on the open edge of the pockets are close to the charged headgroups (Fig. S8a). For the headgroup interacting residues R120 and Q100 in YnaI, there seems no equivalent residues in MscS by just comparing their peptide sequences. However, by comparing the YnaI and MscS structures, we found residues D67 and R59 in MscS that can be superimposed to R120 and Q100, respectively, in YnaI by rotating the whole paddle around the axis of the central channel for ~10° (Fig. S8b). It has been demonstrated in MscS, D67R1 (R1 is a Cys labeled with S-(2,2,5,5-tetramethyl-2,5-dihydro-1H-pyrrol-3-yl)methyl methanesulfonothioate) affects the gating[6] and R59L causes a gain-of-function phenotype[9]. Thus, it is reasonable to speculate that R120 and Q100 in YnaI are important for the lipid sensing in the pocket and thus critical for the channel function. To verify that, we investigated the following mutants Q100A and R120A (Fig. S9). For Q100A, its cell growth was dramatically inhibited when we started to induce the mutant. More interestingly, no protein expression in the remaining cells could be detected by western blot. These results suggest that Q100A is extremely toxic, probably trapping the YnaI channel in the open state. For R120A, the cell growth rate was reduced, but the protein expression level of the mutant was comparable to the wild type. These results suggest that R120A mutant leaks but does not kill the cells, which is similar to the gain-of-function mutation observed in MscS[9]. This is further confirmed by the result of osmotic downshock experiment that R120A protects E. coli more efficiently than wild type YnaI (Fig. S9).

**Ion-conducting pathway**. We submitted our final YnaI model to MOLEonline, a web-based tool to analyze channels and pores within biomolecules[25], and calculated its possible conducting pathway using a bottleneck radius of 0.5 Å (Fig. 4a). Here, we discuss the pathway beginning from the extracellular side of YnaI, where there is a funnel-shaped hole with its narrow end pointing downwards to the cytosolic side. The funnel roughly occupies the space of the outer leaflet of the membrane bilayer. The conducting channel physically starts within the inner leaflet of the bilayer and further extends into the cytoplasm. It is formed by TM3a (residues 142 L to 160 G) from all seven subunits. The radius of the channel starts from 6–8 Å on the top and narrows down to 3–4 Å. The pore lining residues along the TM3a helices are S143, L146, G150, L154, M158, representing a quite hydrophobic environment. The narrowest portion of the channel is between residues L154 and G160, with L154 and M158 acting as two layers of hydrophobic gates (Fig. 4b), equivalent to the "vapor lock" of L105 and L109 in MscS[26]. In addition, right after the kink in TM3 (G160), the side chain of K161 points straight toward the channel center, defining another bottleneck (Fig. 4b). Although it has been proposed that M158 is the main determinant for the cation selectivity of the YnaI channel[19], its equivalent residue, L109 in MscS, has not been linked to the ion selectivity of the channel. In contrast, K161 could potentially play a role in the cation selectivity of YnaI, considering its strong ability to bind to anions.

Because of the horizontally positioned TM3b, the ion-conducting pathway becomes a vestibule in the cytosolic domains, with its widest radius reaching 16–18 Å. The C-terminal domain

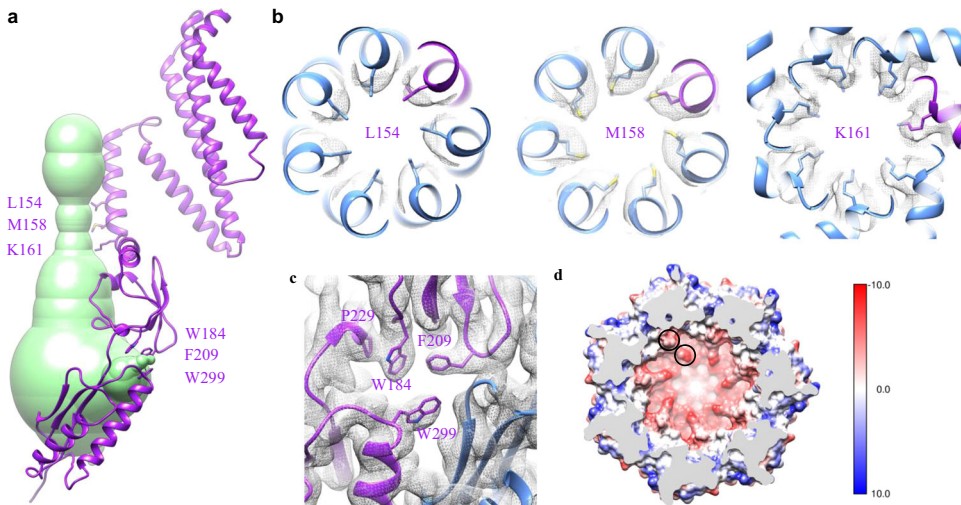

**Fig. 4 Ion-conducting pathway. a** The calculated ion-conducting pathway (green) shown with one subunit of YnaI (purple). Residues at the narrowest positions along the pathway are labeled and shown with side chains. **b**, **c** Zoomed-in views of these critical residues: L154, M158, and K161 rings (top view), as well as the aromatic ones around the side portal (side view), shown with corresponding densities (gray). **d** sliced top view of the cytosolic vestibule colored by coulombic surface charge computed in Chimera using default settings. Q280 and D283 are indicated by black circles.

(equivalent to the β-barrel in MscS) appears to be completely closed, while the side of the vestibule opens seven portals (Fig. 4c). These portals are right in the middle of the β and αβ domains of the same subunit, flanked by the β sheets from an adjacent αβ domain. They are surrounded mostly by flexible loops, which have residues of big side chains, such as W184, F209, and W299. In our model, the distances among W184, P229, and F209 are within 4 Å (Fig. 4c). The ring of P229 interacts strongly with aromatic ring of W184 by the hydrophobic effect and CH-π interaction (between the polarized C-H bonds in P229 and the π aromatic face). The aromatic rings of F209 and W184 stack against each other in a T-shaped fashion. These interactions effectively keep W184 and F209 in position on the side of the portal. It has been reported that W184 is required for function, as even simple substitution to Tyr or Phe makes YnaI inactive[27]. The reason is probably because Trp has the strongest CH-π interaction with Pro[28,29] and mutagenesis of W184 to Tyr or Phe weakens its interaction with P229, which in term makes both aromatic rings of W184 and F209 more flexible. Considering the radius of the side portal is only 3–4 Å, this side chain flexibility could block the portal and thus inactivate YnaI. Interestingly, mutations of the corresponding residue in MscS, Y135, to other aromatic ones retain functionality[27]. In MscS, the only residue interacts with Y135 is F178, equivalent to P229 in YnaI. The interaction is through strong T-shaped π-π stacking with their centroid distance smaller than 5 Å (Fig. S10). It is reasonable that such stacking would still be well maintained when Y135 is mutated to other aromatic residues, and thus does not affect the channel function of MscS[27].

Inside the YnaI vestibule, close to the seven side portals, the surface electrostatic potential appears to be fairly negative (Fig. 4d). Residues contribute to this negative potential include Q280 and D283. It has been suggested that the ion selectivity of MscS-like channels are determined by the residues proximal to the side portals that traps ion to create "ion clouds"[30]. E187 and E227 in *E. coli* MscS, as well as D226 and D229 in *Thermoanaerobacter tengcongensis* MscS[5] are such examples to define the anion selectivity of the channels (Fig. S11). In addition, comparing to the two known MscS channels, the negative surface potential of the whole bottom vestibule in YnaI is even stronger, thanks to these polar residues: S284, S285, S328, Q329, and T339. These observations suggest that cations will likely be trapped in

the vestibule when entered from the side portals, which should make anion transport preferable in YnaI channel. This is contradictory to previously published physiological results[19].

**Heptameric assembly**. To understand how YnaI monomers assemble into a heptamer, we examined the interfaces between the monomers. In the membrane, TM3a from the seven subunits joins together by mostly hydrophobic interactions. We were not able to find specific hydrogen bonds or charged interactions between the TMs, as there are only residues with no or small side chains from the sequence (142-LSGLLTFGGIGGLAVGMAG-160). Below the flat base of TM3b, the interface between two adjacent β domains are through backbone hydrogen bonds between two antiparallelly positioned β strands running side-by-side: 213-PLYVP-217 and 224-ISVEN-228, which effectively connects the neighboring two β sheets (Fig. 5a). In addition, around this region, there are two groups of residues forming complicated side chain interaction networks that contribute to the heptameric assembly: Y215 forms hydrogen bond with R231 and CH-π interaction with P178, as well as N211 forms amino-aromatic interaction with F293 and hydrogen bonds with N234 and Q272 from the adjacent αβ domain (Fig. 5b). The C-terminal domain seems to be another major force for the heptameric assembly, as it appears to be a non-ideal β-barrel (Fig. 5c).

## Discussion

The comparison between YnaI and MscS structures provide valuable insights regarding MscS-like members. In a crystal structure of MscS[6], density of a lipid acyl chain is revealed. In recent MscS nanodisc structures[10,18], lipids are found in multiple locations including the pockets in between paddles (pocket lipids), the central pore (pore lipids), and the periplasmic side near the loop between TM1 and TM2 (hook lipids). While in YnaI, the only visible lipid density is in the pocket. Because our structure is determined in detergent LMNG, it is possible that extra lipids in the pockets and other locations are harshly washed away. Although the associated lipids in MscS are mostly PE and PG[6], it has been confirmed that the channel remains functional in other lipids like PC[31,32]. Thus, it seems that the type of phospholipids is not important for the channel function of MscS. In

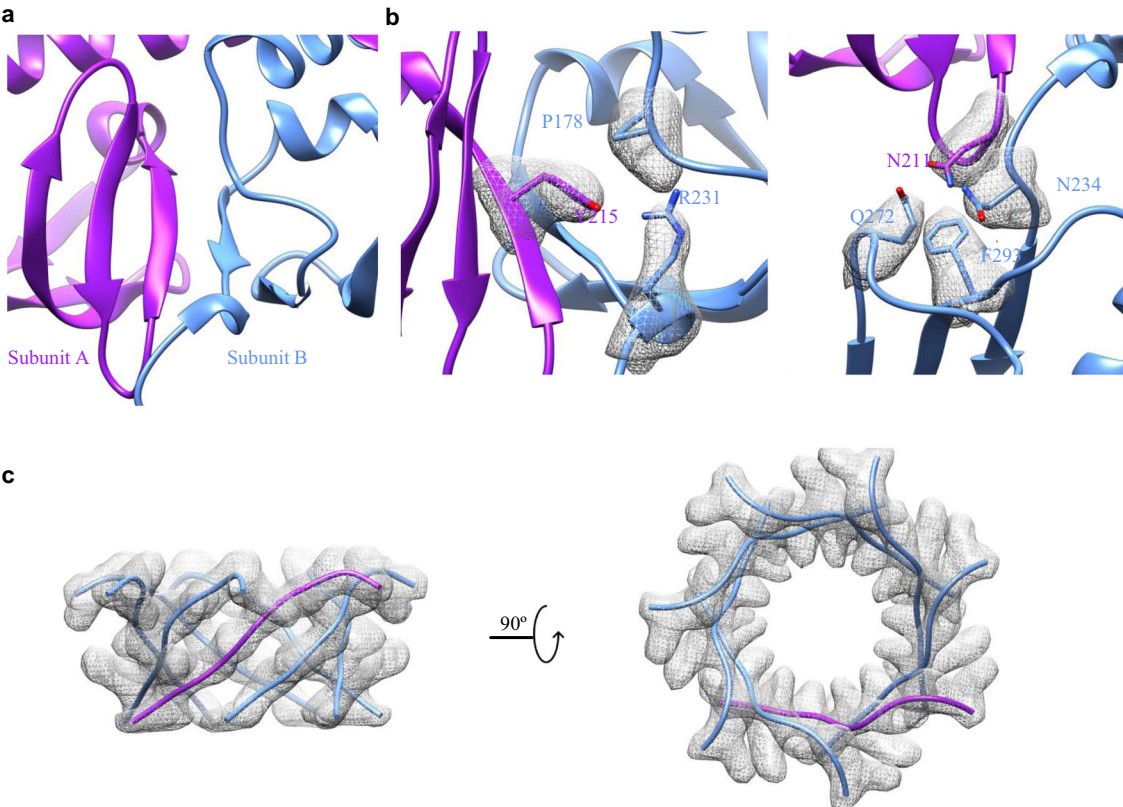

**Fig. 5 Interactions responsible for the heptameric assembly. a** β strands from neighboring β domains interact with each other in an antiparallel fashion. This view is the same view as the β domain shown in Fig. 1a. **b** specific residues in the interfaces between the β domain and its neighboring β/αβ domains. **c** β-barrel like C-terminal domain.

YnaI, all major types of lipids in E. coli membrane are found, with particularly higher percentage of PS and CL. Thus, we argue that, to some extent, YnaI prefers to bind PS and CL. However, the connection between this weak lipid selectivity and mechanosensitive function needs further investigation. Recent structures of MscS have disputed the functional importance of the pocket lipids as their location is too far away from the membrane, and further suggested that the mechanosensing is mediated mainly through the hook lipids instead[10]. However, in YnaI, the pocket lipids are right below the detergent belt, and should be capable of exchanging with lipids in the membrane. Nevertheless, further structural studies of YnaI in lipid environment are required to understand the existence and functional importance of various associated lipids.

Another important insight is that the bigger paddle pockets in YnaI probably bind more lipids, comparing to 2.6–3 lipid molecules estimated for each MscS monomer[33]. A molecular dynamic simulation has shown that as MscS opens, the lipids content decreases by one molecule per pocket[6]. We argue that if the same membrane tension is applied to YnaI, the same amount of pocket lipids will leave and go to the membrane bilayer. However, because of the relatively bigger pocket and more lipid content, the same amount of lipid loss will have a lesser effect on YnaI mechanosensing. Thus, in order to open the YnaI channel, more lipid molecules have to leave the pockets, meaning a higher pressure is required. This is consistent with the idea that more curved membrane area should make YnaI tolerate higher pressure as explained earlier.

With the observations and comparison with known MscS structures, we propose a similar mechanosensing mechanism of YnaI as follows (Fig. 6): at closed resting state without any extra pressure, YnaI sits in the membrane bilayer that is apparently curved because of the tilting of the paddle domains. The pockets in between adjacent paddle domains are filled with mixed types of lipids, including abnormally higher percentage of PS and CL. When the cells experience hypo-osmotic shock, the tension in the membrane bilayer ramps up quickly, physically pulling the curved membrane bilayer away from YnaI. This pulling could have two effects: one is to pull out the lipid molecules in paddle pockets; another is to change the conformation of paddle domains. For the lipids, it is well known that CL promotes negative curvature in the membrane because of its unique conical shape composed of one head group with four lipid tails[34]. Here, by leaving the pockets and joining the membrane, CL may play important role to help decreasing the curvature of the bilayer around YnaI. For the paddles, TM1 and TM2 move as a rigid body in MscS, as they can be superimposed perfectly between open and closed conformations[4]. In YnaI, since the overall interactions between the two helix bundles are vital for the channel stability, we propose that all four helices in the paddle domain move as one rigid body similar to MscS paddles. This paddle movement is probably triggered by the departure of pocket lipids. Thus, these two actions coordinately change the conformation of the YnaI paddles, which in turn pulls the pore lining TM3a outwards and opens the ion-conducting channel. This allows the solutes prefilled in the cytosolic vestibule through side portals to flow out of the cells.

## Methods

**YnaI expression and purification**. The gene encoding full-length YnaI (Uniport: P0AEB5) from *Escherichia coli* (*E. coli*) were cloned into the pET52b (Novagen) vector with a His-tag on the C-termini, which was then over-expressed in *E. coli*

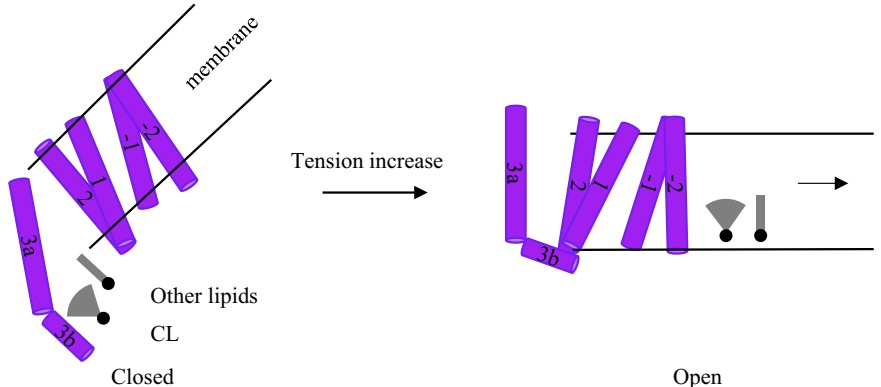

**Fig. 6 Proposed mechanosensing mechanism of YnaI.** When tension increases, the membrane curvature around the closed YnaI decreases, which promotes two actions: lipids leaving the paddle pockets and the paddles switching their conformation. These actions lead to the movement of pore lining TM3a and opens the channel.

strain of BL21(DE3) C43 (Sigma-Aldrich) at 18 °C overnight with 0.2 mM Iso-propyl β-D-1-thiogalactopyranoside (IPTG, UBPBio) in Luria-Bertani (LB) medium. Four-L of harvested cells were resuspended in 20 mM Hepes buffer pH7.5 with 150 mM NaCl, 1 mM phenylmethylsulfonyl fluoride (PMSF), and then lysed by passing through a M110P microfluidizer (Microfluidics) two times at 15,000 psi. The cell lysis was first centrifuged at 15,000 g to get rid of unbroken cells and large cell debris, and then pelleted by ultracentrifugation at 150,000 g for 2 h. The pellets containing cellular membrane fraction were resuspended in the buffer of 20 mM Hepes pH7.5, 150 mM NaCl, and 1 mM PMSF. To solubilize target proteins within the membrane, 1% lauryl maltose neopentyl Glycol (LMNG) was added and the mixture was incubated for 2 h at 4 °C before ultracentrifugation at 150,000 g. The supernatant was collected and incubated with previously buffer-balanced TALON (Clontech) resin. YnaI was eluted with 20 mM Hepes buffer pH7.5, 150 mM NaCl, 200 mM imidazole, 0.01% LMNG, and then incubated overnight in a cold room with thrombin (Enzyme Research Laboratories) at a molar ratio of 1:100 to cut-off the His-tag. YnaI without His-tag was reapplied to fresh TALON resin again and the flow-through were further purified by gel filtration on a Superose 6 column (GE Healthcare Life Sciences). The peak fractions with purified YnaI were concentrated to ~5 mg/ml and stored for further experiments.

**Osmotic downshock experiments.** We generated an *E. coli* strain, namely MS-1106, derived from SIJ488 strain, with the following six MS channels deleted from the genome: MscL, MscS, MscK, MscM, YbdG, and YnaI, by following the published protocol[21]. Then, we introduced the inducible T7 RNA polymerase gene in the strain using the λDE3 lysogenization kit (Millipore Sigma). Using this strain, we performed osmotic downshock assays as previously described with small modifications. MS-1106 (DE3) cells transformed with MscS or YnaI (wild-type and mutants) in pET52b were grown in LB medium with 500 mM NaCl at 37 °C to OD$_{600}$ of 0.6. The cells were induced with 1 mM IPTG for 3 h. To downshock the cells, the culture was normalized to OD$_{600}$ of 0.2, diluted 100-fold into normal LB medium (~170 mM NaCl), incubated at room temperature for 20 min, and then plated as 10 ul drops on agar with a serial of dilutions of 5, 50, 500, and 5000-fold.

**Cell growth curve.** BL21(DE3) C43 *E. coli* cells were transformed with pET52b vectors with various YnaI mutations and grown in LB medium to OD$_{600}$ of 0.6. The culture was diluted to OD$_{600}$ of 0.1 and induced with 1 mM IPTG at 37 °C. The OD$_{600}$ was monitored every 15 min in the next 3 h. The growth curve was prepared in Excel.

**Thin-layer chromatography (TLC).** To get rid of unspecifically associated lipids, purified YnaI samples were cleaned up again with the Superose 6 column. TLC experiment was performed as previously described[35,36]. Briefly, lipids from the samples were extracted with the mixed solution of chloroform and methanol (2:1). TLC plate (Millipore Sigma) was impregnated with 1.8% boric acid in ethanol, air dried and then activated at 100 °C for 30 min. Samples and lipid standards were dotted on the TLC plate. The experiment was performed with a mobile phase of chloroform-ethanol-water-triethylamine (30:35:7:35, v/v/v/v) for 2–3 h. The plate was stained in 50% ethanol with 3.2% sulfuric acid and 0.5% MnCl$_2$ for 20 s, air dried, baked at 120 °C for 30 min, and imaged for analysis.

**Cryo-EM sample preparation and data acquisition.** Three-ul of purified YnaI in LMNG at ~5 mg/ml were mixed with 3 mM fluorinated Fos-Choline-8 (Anatrace) right before applying to a plasma-cleaned C-flat holy carbon grids (1.2/1.3, 400 mesh). The grid was prepared with a Vitrobot Mark IV (Thermo Fisher Scientific) with the environmental chamber set at 100% humidity and 4 °C. It was blotted for

2 s and then flash frozen in liquid ethane cooled by liquid nitrogen. Cryo-EM data were collected on a Talos Arctica (Thermo Fisher Scientific) operated at 200 keV and equipped with a K3 direct detector (Gatan). Totally, 4622 movies were recorded at 45,000x magnification with a calibrated pixel size of 0.864 Å using Leginon[37]. Defocus range was set from −1.2 um to −2.3 um. The parallel illumination set up on the Arctica gave a dose rate of ~24 electrons per pixel per second. Each movie was dose-fractionated to 50 frames with a total exposure of 2 s, leading to a total dose of ~64 electrons/Å$^2$.

**Cryo-EM data processing.** The data were processed using RELION v3.0[16]. Specifically, movies were motion corrected using UCSF MotionCor2[38], and then CTF estimated using Gctf[39]. Approximately 1000 particles were manually picked, and 10 class averages were generated by 2D classification. 5 good class averages were selected as templates for the following auto picking in RELION. Following standard processing of particle extraction, 2D classification, initial model building and 3D classification, best groups of particles were selected and refined to ~5 Å resolution. From there, multiple rounds of CTF refinement and Bayesian polishing were performed. The final resolution was estimated by gold standard FSC to be 3.3 Å. Statistical details can be found in the Table 1.

**Model building, refinement, and validation.** The cryo-EM map used here is at very high quality, allowing de novo model building while using structures of MscS[3,4,6] from E. coli as a visual guidance. The model building was carried out in Coot program[40]. The first transmembrane helix (L5–F30) did not have clear densities to correctly assign side chains, but the backbone was clearly traceable. Thus, we modeled this helix with all Ala residues. In addition, there was no density for the very C-terminal end (N335–R343) of YnaI. The final rounds of model refinement were carried out by real space refinement in PHENIX[41] with secondary structure restraints imposed. The quality of the model was assessed by MolProbity[42]. To validate the refinement, the model was refined against half-maps, and FSC curves were calculated. The final statistics was shown in the Table 1. All figures were prepared using the program UCSF Chimera[43].

**Reporting summary.** Further information on research design is available in the Nature Research Reporting Summary linked to this article.

## Data availability
Cryo-EM map of YnaI was deposited in the Electron Microscopy Data Bank under the accession code EMD-20862. Coordinates of the atomic model of YnaI was deposited in the Protein Data Bank under the accession code 6URT. All other data are available from the corresponding author upon reasonable request.

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

## Acknowledgements

We thank the staff in the shared resources of Biophysics, Mass Spectrometry, and cryo-EM at the University of Colorado Anschutz Medical Campus for their assistance. This work was partially supported by NIH (GM126626 and AG064572).

## Author contributions

H.W., W.Z., and Z.H. designed the experiments, collected and analyzed the data, and wrote the paper.

## Competing interests

The authors declare no competing interests.
