## [Peer Review File · Communications Biology]

Reviewers' Comments:

Reviewer #1:

Remarks to the Author:

This manuscript reports a cryoEM structure of a mechanosensitive channel YnaI, which is homologous to E. Coli MscS. The structure in general is of good quality and reveals five TM helices that are different from the three TM helices in MscS. The authors also present some mutagenesis studies on certain functionally relevant amino acids in the channel. But the functional analyses are superficial without channel recordings, which could provide further insight into mechanical gating mechanisms. Overall the results are of some interest but do not provide significant advance. There are many published MscS structures representing distinct functional states. These published studies have revealed great insight into potential mechanical gating mechanisms in MscS-like channels.

Importantly, a more thorough study on YnaI was recently published in PNAS (Flegler VJ et al. The MscS-like channel YnaI has a gating mechanism based on flexible pore helices. Proc Natl Acad Sci U S A. 2020 Nov 4:202005641. doi: 10.1073/pnas.2005641117.). This PNAS paper reports cryoEM structures of YnaI in the presumed closed and open states in lipid nanodiscs (and electrophysiology experiments), revealing gating transitions of YnaI.

Given the numerous published results in the literature regarding E. Coli MscS and related channels, the current manuscript provides little, if any, significant new insight into mechanical gating mechanisms. In addition, some interpretations in this manuscript are not rigorous and are quite difficult to follow the reasoning. Without rigorous electrophysiology measurements, the current manuscript would be more appropriate to focus on structural comparison with known MscS and YnaI channels.

Major concerns:

1. In 'Introduction', the authors should summarize in more detail the proposed mechanical gating mechanisms from published studies on structural and mechanistic studies of E. Coli MscS and related channels including YnaI. In the current manuscript, there is essentially no description of how MscS gates by membrane tension.
2. While the authors write in length that the membrane domain is tilted and curved in the membrane. Fig.1 indicates a flat membrane boundary, which is probably inspired by the proposed new location for the lipid bilayer (Reddy et al., Molecular basis of force-from-lipids gating in the mechanosensitive channel MscS, *elife* 2019) that is shifted $\sim 14^\circ$ from earlier consensus estimate. Under 'Extended paddle domain', the authors state that 'The elevation and extension of TM1~2 makes the area of the curved lipid bilayer around YnaI much larger than that around MscS. This larger curved area may contribute to the fact that YnaI channel opens at approximately 1.5 times higher pressure comparing to MscS .' This statement regarding the curved area and gating pressure threshold is completely invalid.
3. The 'lipid pockets' described in this manuscript have been previously shown to bind lipid molecules in E. Coli MscS. There are also many empty spaces between paddles, which could bind additional lipids that were not retained in protein purification or not resolved in the structure as these lipids could be weakly bound in the pockets. Therefore, this observation alone does not provide insight regarding mechanical gating in YnaI channels. Thus, the 'Lipid pockets in between the paddles' section does not provide much mechanistic insight.
4. The last paragraph in Discussion. The authors discuss in length about the correlation between 'bigger paddle pockets' and gating pressure threshold, which is not supported by data.

Reviewer #2:

Remarks to the Author:

In the manuscript "Cryo-EM Structure of a MscS-like Channel" the authors have solved a structure at 3.3 Å of YnaI. Additionally they provide supporting experimental evidence for the role of the paddle domain in channel gating in response to mechanical tension. The presence of a potential pocket lipid, suggests that lipids migrate during gating. In general, the bacterial mechanosensitive channel community will be very interested in this structure as it shows many similarities with the recently solved Cryo-EM structures of E. coli MscS.

Overall, I found the structure of YnaI to be interesting and the discussion of the role of the paddle domain in gating to be a good step forward in investigating how bacterial mechanosensitive channels respond to tension. The role of pocket lipid during channel gating is preliminary. While patch clamp experiments are outside the scope of this manuscript I encourage the authors to utilize bacterial patch clamp experiments to further explore how the point mutations have altered the gating of YnaI. Below are specific comments that will strengthen and clarify the manuscript:

1- In the osmotic downshock of YnaI (Figure S6), can you include either WT MscS or WT MscL as a positive control, in both the discussion and the figure.

2- As YnaI was purified using LNMG and the pocket lipid density is not well defined, is it plausible that the observed density is the purifying detergent and not a lipid?

3- In the manuscript you postulate that the gating of YnaI is dependent upon the migration of the pocket lipids, however in the discussion it proposes that the inner leaflet of the cell membrane comes in contact with the pocket lipid. Can you calculate the distance between the pocket lipid and the inner leaflet and discuss how this distance is covered during gating?

4- In the section "Lipid pockets between the paddles": At the end of the first paragraph you state that the location of the lipids is similar to those in MscS (the last sentence in the paragraph), what is the reference to these lipids in MscS?

5- Figures: In general I found the figures to be clear, the suggestions below will increase the clarity of the structural figures.

A- Figure 1: are the dashed lines the predicted location of the lipid head groups?

B- Figure 3: if it doesn't detract from panel A, a box around the region observed in panel B would assist the reader. Additionally, could the predicted location of the lipids be included in the panel A.

C- Figure 5: I am uncertain which region of the channel is being highlighted in panels A and B, perhaps you could include a full length structure of the channel to orient the reader.

D- Figure S5: It is unclear if the TM domains are in the same orientation as they would be in the channel, perhaps you could include a channel monomer to orient the reader. Additionally, the predicted locations of the lipid head groups would be helpful.

Reviewer #3:

Remarks to the Author:

In this manuscript, the authors describe the cryo-EM structure of a bacterial MscS-like channel, named YnaI, in closed state. Moreover, they describe its extended paddle domain that is attached to the N-terminus of MscS, its lipid pockets between paddles, its ion-conducting pathway, and its heptameric assembly based on the cryo-EM structure of YnaI, and discuss its mechanosensing mechanism.

The cryo-EM structure of YnaI in both open and closed states has recently been reported in a PNAS article (vol. 117, 28754–28762) published by Flegler et al. on Nov. 17, 2020, in which the cryo-EM structure of YbiO, which is one of the MscS-like channels, is also reported. Flegler et al. compare three MscS-like channels—MscS, YnaI, and YbiO—and describe the activation mechanism of YnaI and the role of lipids in the pocket between paddles in YnaI, and also characterize the pores and ion-conducting pathways of those three channels.

The comparison of the data described in the present manuscript with that reported by Flegler et al. shows overall similarities in YnaI architecture among the two studies; the two studies also share many topics. Nonetheless, the two studies have different depth analyses and discussion, and some data are controversial, depending on the topic, such as the lipid species that can be accommodated in the pocket and the mechanism of lipid recognition in the pocket, the ion-conducting pathway and mechanism of ion selectivity, and the interaction responsible for the subunit assembly. Most of this information may be relevant for a better understanding of the underlying molecular processes involved in the mechanosensing and gating of mechanosensitive channels.

Mechanosensing is becoming one of the most important topics in biology as cells have to sense different mechanical stimuli affecting biological processes, ranging from cellular osmoregulation in bacteria to hearing and touch, as well as cell proliferation and morphological changes.

Therefore, molecular understanding of the mechanosensing and gating processes of mechanosensitive channels is important, and I consider this manuscript worthy of publication.

However, I still have some concerns that should be addressed by the authors before the manuscript is accepted for publication.

- (1) Title: The title should be revised to express the content in a more specific way.
- (2) Page 6, line 4: The authors should discuss the reason why deltaN70 could not be expressed.
- (3) Figure S6 and related descriptions in the text: Electrophysiological analysis should be performed in mutagenesis studies to elucidate the effects of each mutation on channel activity.
- (4) Page 6, line 13, and many other places throughout the text: Bibliographic references are required to support the described comparison with MscS.
- (5) Figure 3 and related descriptions in the text: Electrophysiological analysis should be performed to verify the involvement of amino acid residues in the recognition of lipids.
- (6) Figure S9b and page 9, lines 11–13: The result of osmotic downshock of wild type should be represented.
- (7) Figure 4 and related descriptions in the text: It would be better to perform mutagenesis studies of Q280 and/or D283 to verify the hypothesis regarding their involvement in ion selectivity.
- (8) Page 10, line 13: “Figure 4b” is mislabeled and it should be “Figure 4c.”

February 11, 2021

Dear Reviewers,

Thanks for the kind comments and constructive suggestions. We understand that electrophysiology experiment is important to understand functions of mechanosensitive channels and would love to do it if possible. This manuscript has been reviewed several times since Jan 2020 with different journals. Every time, electrophysiology experiment was requested. Every time, we tried to find collaborators. Despite extensive efforts, the pandemic makes it very difficult for us to find a reliable collaboration, especially as a newbie in the field. However, we are confident that our biochemical experiments are thorough and should provide an excellent alternative to the electrophysiology experiment that everyone desires. In this letter, we have addressed every comment except about the electrophysiology. Please find our point-to-point response below.

R1's points:

1. In 'Introduction', the authors should summarize in more detail the proposed mechanical gating mechanisms from published studies on structural and mechanistic studies of E. Coli MscS and related channels including YnaI. In the current manuscript, there is essentially no description of how MscS gates by membrane tension.
Response: Agree. We have added more description for the MscS gating mechanism in the Introduction section (highlighted in yellow).

2. While the authors write in length that the membrane domain is tilted and curved in the membrane. Fig.1 indicates a flat membrane boundary, which is probably inspired by the proposed new location for the lipid bilayer (Reddy et al., Molecular basis of force-from-lipids gating in the mechanosensitive channel MscS, *elife* 2019) that is shifted $\sim 14 \text{ \AA}$ from earlier consensus estimate. Under 'Extended paddle domain', the authors state that 'The elevation and extension of TM1~2 makes the area of the curved lipid bilayer around YnaI much larger than that around MscS. This larger curved area may contribute to the fact that YnaI channel opens at approximately 1.5 times higher pressure comparing to MscS .' This statement regarding the curved area and gating pressure threshold is completely invalid.

Response: The boundary of the membrane is an indication of where the edge of the detergent belt is in our cryo-EM map. This does not mean the membrane bilayer is flat. As for the relationship between the curved area and gating pressure, we used the word "may" to give a possible explanation why YnaI needs 1.5x higher pressure to open comparing to MscS. After all, the two proteins are very close to each other and the major difference seems to be the size of the paddle domain, which relates to the size of the curved membrane area. Thus, we reasoned that the larger area might be related to the higher pressure. Of course, we do not have direct evidence for that. However, there is an experiment to test the idea: generate YnaI mutants with more (or less) TM bundles and then measure the opening pressure. Unfortunately, we do not have the electrophysiology resource to do it and would like to pursue it in the future.

3. The 'lipid pockets' described in this manuscript have been previously shown to bind lipid molecules in E. Coli MscS. There are also many empty spaces between paddles, which could bind additional lipids that were not retained in protein purification or not resolved in the structure as these lipids could be weakly bound in the pockets. Therefore, this observation alone does not provide insight regarding mechanical gating in YnaI channels. Thus, the 'Lipid pockets in between the paddles' section does not provide much mechanistic insight.

Response: The reviewer is absolutely right about additional lipids in the pocket. As a matter of fact, the most recent paper published on Feb 10, 2021 (Zhang et al., Nature, <https://doi.org/10.1038/s41586-021-03196-w>) reveals multiple pocket lipids in MscS. The merit of this section in our manuscript is the identification of residues potentially interacting with lipid headgroups, which are functionally important as demonstrated by our experiments. Importantly, the lipid-binding residues we identified here are different from a previous publication (Flegler et al., PNAS 2020). This makes perfect sense because there should be multiple lipids in the pocket, and they shall require multiple residues for the interactions.

4. The last paragraph in Discussion. The authors discuss in length about the correlation between ‘bigger paddle pockets’ and gating pressure threshold, which is not supported by data.

Response: This comment is similar to the point #2. We appreciate the reviewer’s concern about this idea. By comparing YnaI and MscS, we know that YnaI has a higher gating pressure and bigger paddle pockets. We are trying to make sense of these facts by providing a possible explanation using logical reasoning. We understand that we could be totally wrong when new findings are revealed. But, at this point, we are confident about this idea.

R2’s points:

1- In the osmotic downshock of YnaI (Figure S6), can you include either WT MscS or WT MscL as a positive control, in both the discussion and the figure.

Response: Thanks for the suggestion. We included WT MscS as a positive control as it is in the same family as YnaI. The corresponding figure and description are both updated.

2. As YnaI was purified using LMNG and the pocket lipid density is not well defined, is it plausible that the observed density is the purifying detergent and not a lipid?

Response: With both the MS and TLC experiments, we are certain that there are lipids bound to YnaI. However, we do not observe any lipids-like densities in locations other than the pockets. In addition, in FigS7, the R120A mutant binds less lipids comparing to the WT, suggesting that R120 may interact with lipid. While the chemical nature of detergent LMNG makes it much less likely to interact with R residues. Thus, it is most likely that the pocket density is lipid, not LMNG.

3. Can you calculate the distance between the pocket lipid and the inner leaflet and discuss how this distance is covered during gating?

Response: The vertical distance between the presumed pocket lipid headgroup and the detergent belt in our closed YnaI map is only $\sim 8\text{\AA}$. The hypothesis is that when the channel opens, the tension in the bilayer moves the helices and the pocket will be buried within inner leaflet. It is supported by multiple R and W residues found at the bottom of the pocket, which can be the lipid headgroup anchors in the open state. In addition, the most recent MscS paper (Zhang et al., Nature, <https://doi.org/10.1038/s41586-021-03196-w>) echoes the same idea that lipids in the solute-exposed pocket are pulled out during conformational transition from closed to open.

4. In the section “Lipid pockets between the paddles”: At the end of the first paragraph, you state that the location of the lipids is similar to those in MscS (the last sentence in the paragraph), what is the reference to these lipids in MscS?

Response: Reference added. (highlighted in yellow)

5A: Figure 1: are the dashed lines the predicted location of the lipid head groups?

Response: Yes, they match the edges of the detergent belt in the cryo-EM map.

5B: Figure 3: if it doesn’t detract from panel A, a box around the region observed in panel B would assist the reader. Additionally, could the predicted location of the lipids be included in the panel A.

Response: The figure is remade as suggested.

5C: Figure 5: I am uncertain which region of the channel is being highlighted in panels A and B, perhaps you could include a full-length structure of the channel to orient the reader.

Response: Sorry the full-length structure is probably too big for a panel. But we now add sentence to orient the readers: “This view is the same view as the β domain shown in Figure 1a.”

5D: Figure S5: It is unclear if the TM domains are in the same orientation as they would be in the channel,

perhaps you could include a channel monomer to orient the reader. Additionally, the predicted locations of the lipid head groups would be helpful.

Response: The figure is remade as suggested.

R3's points:

1. Title: The title should be revised to express the content in a more specific way

Response: We have changed the title to be: "Extended Sensor Paddles with Bound Lipids Revealed in Mechanosensitive Channel YnaI".

2: Page 6, line 4: The authors should discuss the reason why deltaN70 could not be expressed.

Response: By deleting the first two transmembrane helices, it is most likely that the folding of transmembrane domain is completely messed up and so it cannot be inserted into the membrane as WT.

3: Figure S6 and related descriptions in the text: Electrophysiological analysis should be performed in mutagenesis studies to elucidate the effects of each mutation on channel activity.

Response: As mentioned in the beginning of this letter, we would love to do it but cannot. In addition, we believe the biochemical experiments described are good alternatives to the patch-clamp.

4: Page 6, line 13, and many other places throughout the text: Bibliographic references are required to support the described comparison with MscS

Response: Although most MscS references are mentioned in page 4, sorry for missing it in specific places. We have added the references as suggested in many places (highlighted in yellow).

5. Figure 3 and related descriptions in the text: Electrophysiological analysis should be performed to verify the involvement of amino acid residues in the recognition of lipids.

Response: Same as for point #3.

6. Figure S9b and page 9, lines 11–13: The result of osmotic downshock of wild type should be represented.

Response: We have added the WT result into Figure S9b.

7. Figure 4 and related descriptions in the text: It would be better to perform mutagenesis studies of Q280 and/or D283 to verify the hypothesis regarding their involvement in ion selectivity.

Response: As mentioned before, we cannot perform electrophysiology experiments. The reviewer raised an important point about selectivity, and this is exactly what we want to study in the future.

8. Page 10, line 13: "Figure 4b" is mislabeled and it should be "Figure 4c."

Response: It is now corrected.

Thanks for your consideration and we are looking forward to hearing back from you.

Sincerely yours,

Reviewers' Comments:

Reviewer #1:

Remarks to the Author:

My major concerns remain as the authors essentially ignored my previous criticisms (Points 2 and 4) regarding the correlation of the curved area and gating pressure threshold and the correlation of 'bigger paddle pockets' and gating pressure threshold. These claims are not supported by any data and are most likely incorrect from studies of other mechanosensitive channels with much more curved areas such as Piezo channels.

As also suggested by another reviewer (which I strongly agree), rigorous electrophysiology experiments to measure single-channel conductance and/or gating pressure threshold are necessary to support mechanistic interpretations presented in this manuscript.

As I pointed out previously and also noted by another reviewer, a more thorough study on YnaI was recently published in PNAS (Flegler VJ et al. The MscS-like channel YnaI has a gating mechanism based on flexible pore helices. Proc Natl Acad Sci U S A. 2020 Nov 4:202005641. doi: 10.1073/pnas.2005641117.). This PNAS paper reports cryoEM structures of YnaI in the presumed closed and open states in lipid nanodiscs (together with electrophysiology experiments), revealing gating transitions of YnaI. It is very surprising that the authors did not even mention the above thorough and directly related work in the revised manuscript.

Reviewer #2:

Remarks to the Author:

The authors have considered my comments and the manuscript and figures are much clearer.

Reviewer #3:

Remarks to the Author:

This manuscript has been improved according to suggestions of reviewers except for electrophysiological experiments.

I think that the electrophysiological method is very useful for the analysis of mechanosensitive channels and cannot be completely replaced by biochemical experiments.

Although COVID-19 is an obstacle to research, I disagree with making COVID-19 a reason to tolerate a decline in the quality of science.

However, this manuscript contains sufficient scientific results that provide important findings about Msc which would lead to elucidate the lipid species that can be accommodated in the pocket, the mechanism of lipid recognition in the pocket, the ion-conducting pathway, mechanism of ion selectivity, and the interaction responsible for the subunit assembly.

This manuscript contains a deeper and more detailed analysis of YnaI compared to the previous paper published in PNAS.

Electrophysiological experiments provide important insights into Msc that cannot be completely replaced by biochemical experiments, but the results obtained do not provide a complete understanding of the physiological role.

Physiological experiments are needed to understand how the characteristics of Msc obtained from structural analysis and electrophysiological experiments are related to the physiological functions of Msc.

In this manuscript, the authors express the mutant Msc (YnaI) created based on the considerations obtained from the structural analysis in living cells and verify the considerations based on the structural analysis.

Therefore, I think it is worth publishing, even if it does not include electrophysiological experiments.

Specifically, the following points.

- 1) The importance of interaction between the two bundles is being verified by expressing Msc (YnaI) in which amino acid residues that are supposed to be involved are substituted in living cells.
- 2) The lipid pocket is described in more detail, and Msc (YnaI) substituted with an amino acid residue, which is considered to be important for the recognition of lipid molecules, is expressed and verified in living cells.
- 3) The types of lipids that bind to Msc (YnaI) have been confirmed by biochemical experiments.
- 4) The ion-conducting pathway is analyzed in more detail, and important considerations have been made.
- 5) There is no description of the multimer formation mechanism in the paper published in PNAS, but this manuscript has been analyzed.

Some of the above are inconsistent with the findings so far, but these may also contribute to the understanding of Msc.

The new points I noticed are as follows.

Line 181: The description in the text is "the loop between TM-2 and TM-1", but in Figure 3b it is "TM1-2 loop". Please fix it to the correct one.

Figure S7: Lane numbers 8 and 9 should be noted.
The result of R120A should be described in the text.

Figure S9b: The difference between WT and R120A should be tested for statistically significant differences.

March 14, 2021

Editorial Office
Communications Biology

Dear Reviewers,

Thank you for your help along the way. In this revision, we have made changes according to the comments (all highlighted in the manuscript). Our point-to-point responses are as follows:

Reviewer #3:

Line 181: The description in the text is "the loop between TM-2 and TM-1", but in Figure 3b it is "TM1-2 loop". Please fix it to the correct one.

Response: The text in Fig 3b was wrong. It's now fixed.

Figure S7: Lane numbers 8 and 9 should be noted. The result of R120A should be described in the text.

Response: Lane numbers are changed as suggested. A sentence about R120A result is added in page 8.

Figure S9b: The difference between WT and R120A should be tested for statistically significant differences.

Response: A T-test was performed, and the corresponding p-value is now added in the figure.